# Factors Influencing Public Willingness to Reuse the Unused Stored Medications in Jordan: A Cross-Sectional Study

**DOI:** 10.3390/healthcare11010075

**Published:** 2022-12-27

**Authors:** Hamza Alhamad, Deema Jaber, Rana Abu-Farha, Fares Albahar, Sahar M. Edaily, Parastou Donyai

**Affiliations:** 1Department of Clinical Pharmacy, Faculty of Pharmacy, Zarqa University, P.O. Box 2000, Zarqa 13110, Jordan; 2Department of Clinical Pharmacy and Therapeutics, Faculty of Pharmacy, Applied Science Private University, Amman 11931, Jordan; 3Department of Pharmacy and Forensic Science, King’s College, London SE1 9NH, UK

**Keywords:** medicine reuse, medicine waste, unused medicines, public willingness, medicines storage, dosage form, sustainability, waste management, pharmacological class

## Abstract

The reissuing, redispensing, redistributing, or even recycling of the returned unused stored medicines is referred to as medicine reuse. A few studies have examined people’s willingness to reuse unused stored medicines. This study aims primarily to explore the public’s willingness to reuse unused stored medicines in the future and the factors affecting their decisions, and to assess the pharmacological types and formulations of the unused stored medicines by the public. A cross-sectional study was conducted using a convenient sampling technique over three weeks (February–March 2022) among Jordanian populations. Data were collected from 681 participants who agreed to fill out a questionnaire, which was distributed online. Participants showed a positive perception of reusing any of the unused stored medicines in the future; of those, 79.6% (*n* = 400) believed that reusing “unused medicines” has economic benefits and 50% (*n* = 251) believe there are environmental benefits. Moreover, more than half of the participants believed that reusing “unused medicines” is safe (*n* = 256, 51%), but provided that a pharmacist or a doctor evaluates their quality or that the unused medicines are stored well. Results from this study show that the public positively endorses the idea of reusing unused stored medicines if the quality and safety are assured. However, future studies are needed to evaluate the Jordanian healthcare professionals’ perceptions towards medicine reuse and pharmaceutical companies’ willingness to be a part or facilitator of medicine reuse.

## 1. Introduction

In Jordan, medicines are obtained from community and hospital pharmacies [1]. In hospital settings, minimal interaction occurs between the pharmacist and the patient. Most hospital pharmacies still have antiquated dispensing windows where medicines are placed for patients to pick up [2]. Admitting that the Jordanian Food and Drug Administration (JFDA) drug classification and law regarding drug dispensing is quite similar to those in the West, these laws are not strictly enforced or followed in the community pharmacies in Jordan [2,3]. At the community pharmacy, the patient can buy any medicine without a prescription, except for misoprostol, sedative-hypnotics, controlled narcotics, and major tranquilizers (e.g., benzodiazepines), which can only be dispensed upon the issue of a special prescription signed by a registered physician [4]. These would all add to the accumulation of unused stored medicines referred to as medicines waste. Medicines waste has environmental and economic impacts, and poses risks to human health [5]. Healthcare professionals in Jordan do not require or even educate people about returning unused stored medicines, and there is scarce evidence about the disposal practices of unused stored medicine in Jordan [6]. A recent Jordanian study reported that a high percentage of the population improperly disposes of their unused stored medications by throwing in general household waste and flushing down the sink or the toilet, which would impact the wildlife and human environments [6]. Although anecdotal observations confirm these practices [1,6,7], there is a lack of evidence or research investigating the environmental impacts of these unfavorable disposal practices of unused medicinal waste in Jordan. The economic impacts of the unused stored medicines are possibly more destructive to the brittle Jordanian economy [7]. A study conducted in 2012 reported that the total extrapolated cost of unused medicines stored in patients’ homes in Jordan was around $30 million [7]. One year later, another study reported that the total cost of unused medicines stored in patients’ homes from Amman (the capital city of Jordan) was $12 million [1]. Lastly, unused stored medicines pose a risk to human health [5,8]. The risk to human health in Jordan was evidenced by two studies [1,9]. One study referred to inappropriate storage conditions, with around 50% of unused medicines stored outside pharmacy cabinets in places considered unsafe, accessible to children, and with the potential risk of accidental poisoning, especially with children [1]. Another study described the high risk of antibiotic resistance as a result of self-medicating unused stored antibiotics [9]. The unused stored medicines’ impact on the environment, economy, and its risk to human health warrant a solution. One potential solution for medicines waste would be reissuing or redispensing the returned unused stored medicines [5,10,11]. The reissuing, redispensing, redistributing, or even recycling of the returned unused stored medicines is referred to as medicine reuse [8,10]. Medicine reuse has different interpretations and definitions. For example, patients reuse their own medicines during hospital admission [12,13,14], recycle unused stored medicines, or repackage for manufacturing processes [10]. Additionally, medicines reuse can be referred to as repurposing unused stored medicines for new diseases or conditions [10]. These definitions and interpretations are important, but exceed the focus of this study. The term “medicine reuse” is utilized in this study in the context of the community pharmacy and was referred to as the reissue of medicines returned unused to the community pharmacies [8]. This study focuses on the Jordanian context, where community pharmacists yet cannot reissue the returned unused stored medicines [2,7]. Stakeholders from different countries have reported concerns about medicine reuse [5,10]. These concerns are about (i.e., but not limited to) the quality and the safety of the returned unused medicines, their storage conditions, the risks of contamination or sabotage, and the practicality and financial worthiness of medicine reuse based on pharmacological types and the pharmaceutical formulations of the unused stored medicines [5,8,10]. Additionally, the technology that would be used to trail the medicine’s storage conditions, and the manufacturing and expiry date to manage resupply should reasonably be cost-effective [15,16]. These concerns are significant and would affect the implementation of medicine reuse as a scheme in Jordan and globally. However, this exceeds our focus in this study. A limited number of global studies have examined people’s views on the concept of medicine reuse [5,11,17,18,19,20]. This study aims primarily to explore the public’s willingness to reuse unused stored medicines in the future and the factors affecting their decisions, as well as to assess the pharmacological types and formulations of the unused stored medicines by the public.

## 2. Materials and Methods

### 2.1. Compliance with Ethical Standards

This study was approved by the University Ethics Committee for Scientific Research (ECSR) (Appendix B). In addition, an informed consent form (Appendix D) was collected from all participants before participation in the study, ensuring voluntary participation and that the participants could withdraw at any stage with their answers treated confidentially.

### 2.2. Study Design and Sample 

This study was conducted using a convenient sampling technique over three weeks (February–March 2022) among Jordanian populations. The cross-sectional study design was used to assess the public’s willingness to reuse unused stored medicines in the future and the factors affecting their decisions, as well as to assess the pharmacological types and formulations of the unused stored medicines by the public. The sample size calculation was determined using the Raosoft sample calculator [21]. Considering the population in Jordan is around 10 million and a half [22], the sample size was calculated by determining a margin of error (5%), confidence level (95%), and response distribution (50%). A sample size of 385 was found to be minimally required. 

### 2.3. Questionnaire Development and Data Collection 

A self-reported questionnaire was developed by the authors in concordance with the research aims and based on previous studies in the literature [5,10,11,17,18,19,20]. The questionnaire was developed at first in English, and then translated into Arabic with the help of a professor in English translation whose native language is Arabic. To ensure the face validity of the questionnaire, both the Arabic and English versions of the questionnaire were reviewed by a panel of four academic experts in pharmacy practice and reviewed accordingly. The academic experts commented on the wording, clarity, and comprehensiveness of the questionnaire items, and whether each item was relevant to the study’s aims and objectives. The academic experts’ feedback and comments were reviewed by the Authors and used to refine the questionnaire. After completing the piloting process, the final questionnaire version was created using Google forms and distributed online based on the authors’ networks via social media platforms: Facebook, WhatsApp, and Twitter. Participants willing to participate were also sent the study ethics committee approval (Appendix B), the consent form (Appendix C), and the questionnaire, which required less than 10 min to complete. The final questionnaire (Appendix D) comprised two main sections. The first section sought to obtain demographic information about the participants. The second section explored the pharmacological types and the pharmaceutical formulations of unused stored medicines, the public’s willingness to reuse stored medications in the future, and the factors affecting their decisions. Appendix A depicts a flowchart of the whole study process.

### 2.4. Statistical Analyses

The statistical package for social science (SPSS) version 22 (SPSS Inc., Chicago, IL, USA) was used to analyze the data collected in this study. Frequency/percentage was utilized for qualitative variables. The study was conducted to determine predictors of the participants’ willingness to reuse unused stored medications. It was hypothesized that age, gender, marital status, education level, profession, outcome level, the presence of insurance, and taking chronic medications could predict the public’s willingness to reuse stored medications. To test this hypothesis, a logistic regression analysis was conducted. Following a simple logistic regression, any variable with a *p*-value < 0.250 was considered eligible for entry into the multiple logistic regression analysis. All variables were checked for any absence of multicollinearity before performing the multiple logistic regression analysis (i.e., Pearson correlation coefficient < 0.9 for any two variables). A *p*-value (≤0.05) was considered statistically significant when identifying factors affecting participants’ willingness to reuse unused stored medicines. The validity of the multiple logistic regression model was assessed using Nagelkerke R2 and the application of the Hosmer–Lemeshow goodness-of-fit test, where a *p*-value > 0.05 indicated that the model fits with the data.

## 3. Results

During the study period, 681 participants agreed to participate in this study. More than half of the surveyed participants’ age are 40 years or lower (*n* = 378, 58.4%). Additionally, around two-thirds are females (*n* = 411, 60.4%) and 61.5% are married (*n* = 419). Regarding the educational level, around two-thirds hold a university degree (*n* = 454, 66.7%) and more than 41% of them reside in Amman (*n* = 281, 41.3%). Furthermore, 61.7% of the participants have a medical-related degree (*n* = 420) and around one-third (*n* = 215, 31.6%) have an income of less than 500 Jordanian dinars/month. Finally, around 41% of the participants (*n* = 218, 41.3%) are non-insured. For more details about the participants’ sociodemographic characteristics, refer to Table 1.

Table 2 summarizes the participants’ medication-related information. Around 29% of the respondents (*n* = 194, 28.5%) reported taking medications for more than six months. Moreover, all the participants reported that they have unused stored medications (*n* = 681, 100%), with around 59% of them (*n* = 404, 59.3%) reporting checking the expiration date regularly. 

Concerning the unused stored medications’ pharmacological classes (Figure 1), the main reported pharmacological types of unused stored medication were pain-relieving medications (*n* = 407, 59.8%), followed by medicines to treat colds, coughs, and influenza (*n* = 397, 58.3%), and antibiotics (*n* = 292, 42.9%). At the same time, the most common stored pharmaceutical formulations (Figure 2) were oral medications, such as capsules and pills (*n* = 539, 79.1%), followed by creams and ointments (*n* = 437, 64.2%). The least stored pharmaceutical formulation was the parenteral formulation, such as intradermal, intramuscular, and intravenous medications (*n* = 134, 19.7%).

Participants were asked to identify their reasons for storing unused medicines as reported by the study participants (Table 3). Results showed that more than 65% of them stored medications for future use (*n* = 448, 65.8%), while 38.5% of them (*n* = 262) reported storing unused medicines to donate to family members, friends, or anyone who needs them. Moreover, 24.5% of the participants (*n* = 165) reported that they store unused medicines because they are redundant. Finally, only 8.5% of the participants (*n* = 58) revealed that they store unused medicines because they do not know how to dispose of them. 

Participants were asked about their willingness to reuse any unused stored medications in the future, and around three-quarters of the participants revealed that they are willing to reuse any unused stored medications in the future (*n* = 502, 73.7%). Those willing to reuse medications were asked to report perceptions toward reusing medicine (Table 4). The results show that 79.6% of the participants believe that reusing “unused medicines” has economic benefits, as it reduces the costs of treatment to the individual and the country. In addition, 54.8% (*n* = 273) and 51.0% (*n* = 256) believe that reusing “unused medications” is safe, but provided that a pharmacist or a doctor evaluates the quality of them or that the unused medicines are stored well, respectively. Furthermore, half of the participants (*n* = 251) believe that reusing “unused medicines” has environmental benefits, as it reduces the environmental impacts caused by malpractice related to drug disposal.

Finally, regarding predictors of participants’ willingness to reuse unused stored medications (Table 5), results reveal that female participants (Odds Ratio = 0.616, *p* = 0.013) and those with higher income (≥500 Jordanian dinars) (Odds Ratio = 0.451, *p* = 0.002) show a lower willingness to reuse unused stored medications. Moreover, those who have medical professions are less willing to reuse unused stored medications (Odds Ratio = 1640, *p* = 0.011). The multiple logistic regression model fits well, χ^2^(8) = 9.906, *p* = 0.272 (Hosmer–Lemeshow goodness-of-fit). The model explains 6.7% (Nagelkerke R^2^) of participants’ willingness to reuse unused stored medicines and correctly classifies 74.8% of cases.

## 4. Discussion

Medicine reuse originates as a solution to medicine waste; therefore, if there is zero waste, there will be no medicine to be reused. With a growing number of studies highlighting the idea of reusing medicines [5,10,11,17,19,20,23], this study contributes by exploring the Jordanian public’s willingness to reuse unused stored medicines in the future and factors affecting their decisions. In this study, around one-third of the participants reported having medicines for more than six months. Additionally, all participants reported that they have unused stored medications, with more than half regularly checking their medicines’ expiry dates. This was consistent with results from a recent Jordanian study where about half of the participants have between one and five unused stored medicines at home [6]. In addition, previous evidence from two Jordanian studies reported that people stored unused medicines at home [1,7]. Concerning the unused stored medicines’ pharmacological classes, this study’s main reported pharmacological type was pain-relieving medications, followed by medicines to treat colds, coughs, influenza, and antibiotics. Additionally, the most commonly stored pharmaceutical formulation was oral dosage forms, such as capsules and pills, followed by creams and ointments. The least stored pharmaceutical formulation was the parenteral formulation, such as intradermal, intramuscular, and intravenous medications. This is consistent with results from a recent narrative review from different countries that reported pain-relieving medications, followed by medicines to treat colds, coughs, influenza, and antibiotics, to be a common pharmacological type of unused stored medication [8]. Additionally, results from previous Jordanian studies [1,6,7] are consistent with the results from our study. Lastly, oral medications were reported to be the most common pharmaceutical formulation in many studies from different countries [6,24,25,26,27,28,29,30,31,32,33,34,35,36,37,38,39]. This is important as oral pharmaceutical formulations are more suitable for reuse [11] compared to other dosage forms, such as liquids or injectables, taking into consideration other factors to be checked by pharmacists before medicines are reused, such as safety, quality, storage conditions, being sealed, unopened, non-tampering, and not contaminated [5,8,10,11,17]. Findings from this study demonstrate that more than half of the participants stored unused medicines for future use, while more than one-third reported storing unused medicines to donate to family members, friends, or anyone who needs them. Additionally, a quarter of the participants reported that they store unused medicines because they are redundant, with less than 10% revealing that they store unused medicines because they do not know how to dispose of them. Around three-quarters of the participants revealed that they are willing to reuse any unused stored medicines in the future; of those, more than three-quarters of the participants believe that reusing “unused medicines” has economic benefits and half of the participants believe there are environmental benefits. Additionally, more than half of the participants believe that reusing “unused medicines” is safe, but provided that a pharmacist or a doctor evaluates their quality or that the unused medicines are stored well. These are consistent with results from other studies [5,10,11,15,17,19,20]. Lastly, female participants and those with higher incomes show a lower willingness to reuse unused stored medicines. This would be explained as both females and higher income individuals would like to have brand new rather than reusing unused stored medicines [17,40]. Additionally, those who have medical professions are less willing to reuse unused stored medications. This would be significant considering the need to assess their perceptions of reusing unused stored medicines and factors that affect their decisions. 

This is the first Jordanian study highlighting the public’s willingness to reuse unused stored medicines. The current study should be considered in light of some limitations. First, data were collected online; hence, only people who use the Internet and other social media platforms were able to participate. Moreover, all information in this study was obtained through the self-report method with a risk of social desirability bias or recall bias. Furthermore, we used a convenience sampling method that would not represent the whole Jordanian population. Findings from this study show that most people have positive perceptions and willingness toward medicines reuse if the safety and quality assurance of reissued medicines can be assured. Despite the positive public willingness to reuse unused stored medicines, it is important to clarify that the public decisions to reuse unused stored medicines would be affected by factors related to attitudes (e.g., good or bad, harmful or beneficial, and worthless or worthwhile), social norms (e.g., public values, opinions, and thoughts), and control factors (e.g., sealed medicines, medicines quality, and safety checking), which were not completely explored in this study [5,41]. This would be another limitation of this study. The results from this study would help policy decision-makers in Jordan and comparable global countries gain more insight into the idea of medicine reuse, thus allowing them to react by defining the barriers and facilitators to medicine reuse, and updating their regulations if medicine reuse becomes a reality.

## 5. Conclusions

Medicine waste as a problem has been identified for decades, and there is good evidence that medicine reuse would be a great idea to solve medicine waste. The prevalence of the public’s willingness to reuse unused stored medications and the important factors affecting their decisions are evidenced in the literature in different countries that have reported positive perceptions toward reusing unused stored medicines considering quality and safety assurance. Findings from this study indicate that the public positively endorsed the idea of reusing unused stored medicines if quality and safety are assured. Yet, future studies are required to assess the Jordanian healthcare professional’s perceptions toward medicine reuse and the pharmaceutical companies’ willingness to be a part or facilitator of medicine reuse.

## Figures and Tables

**Figure 1 healthcare-11-00075-f001:**
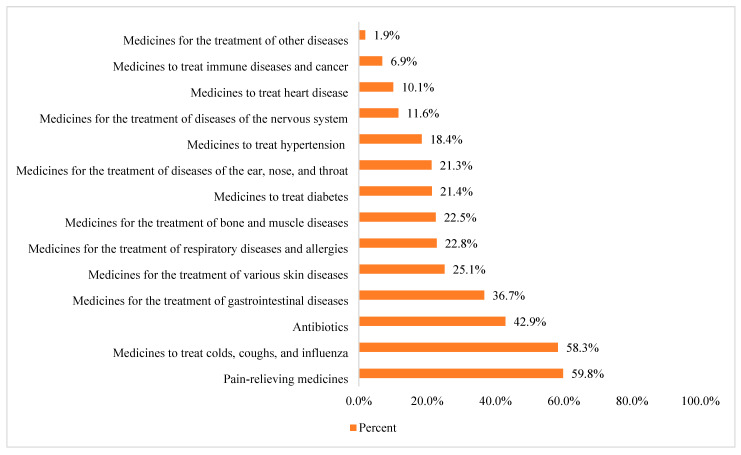
Pharmacological types of unused medicines by study participants (note: participants could select more than one option).

**Figure 2 healthcare-11-00075-f002:**
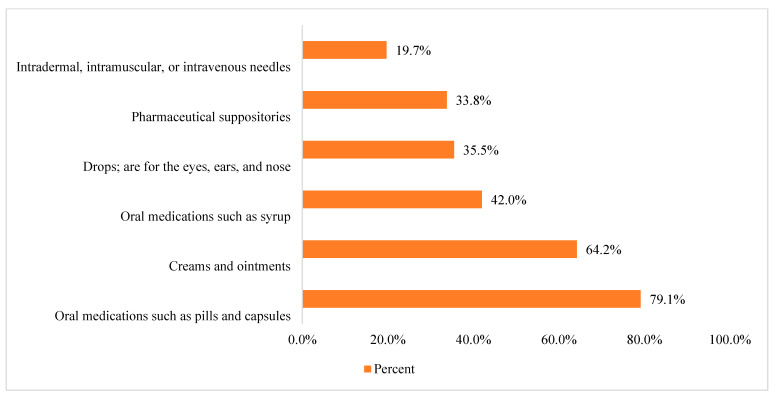
Pharmaceutical formulations of unused medicines by study participants (note: participants could select more than one option).

**Table 1 healthcare-11-00075-t001:** Sociodemographic characteristics of the study participants (*n* = 681).

Parameters	*n* (%)
Age	
<25 years	216 (31.7)
26–40 years	182 (26.7)
41–55 years	190 (27.9)
≥56 years	93 (13.7)
Gender	
Male	270 (39.6)
Female	411 (60.4)
Marital status	
Married	419 (61.5)
Others (single, widowed, or divorced)	262 (38.5)
Level of education	
High school or less	34 (5.0)
Diploma or Bachelor’s degree	454 (66.7)
Postgraduate studies (Master’s or Ph.D.)	193 (28.3)
Place of residence (city)	
Irbid	268 (39.4)
Amman	281 (41.3)
Zarqa	120 (17.6)
Other Cities	12 (1.8)
Profession	
Non-medical profession	261 (38.3)
Medical profession ^1^	420 (61.7)
Income level	
<500 Jordanian dinars ^2^	215 (31.6)
500–1000 Jordanian dinars	241 (35.4)
1001–1500 Jordanian dinars	103 (15.1)
>1500 Jordanian dinars	116 (17.0)
Missing data	6 (0.9)
Insurance	
Not insured	281 (41.3)
Insured	460 (58.8)

^1^: e.g., physician, dentist, pharmacist, nurse, ^2^: Jordanian Dinar = 1.41 United States Dollar.

**Table 2 healthcare-11-00075-t002:** Medication-related information of the study participants (*n* = 681).

Parameters	*n* (%)
Are you currently taking medication for a long time (more than six months)?	
No	487 (71.5)
Yes	194 (28.5)
Do you have any unused stored medicine?	
Yes	681 (100%)
No	0 (0.0)
Do you check the expiration date of unused medication?	
Yes, regularly	404 (59.3)
Yes, often	209 (30.7)
Yes, rarely	61 (9.0)
No	7 (1.0)

**Table 3 healthcare-11-00075-t003:** Reasons for storing unused medicines as reported by the study participants (*n* = 681).

Parameters	Percent Agreed
	*n* (%)
I store unused medicines for future use	448 (65.8)
I store unused medicines to donate to family members, friends, or anyone who needs them	262 (38.5)
I store unused medicines because I do not know how to dispose of them	58 (8.5)
I store unused medicines because they are redundant	165 (24.2)

**Table 4 healthcare-11-00075-t004:** Perceptions toward reusing medicines among those participants who are willing to reuse medications (*n* = 502).

Reasons	Percentage Agreed
	*n* (%)
Reusing “unused medicines” has economic benefits, as it reduces the costs of treatment to the individual and the country	400 (79.6)
Reusing “unused medicines” has environmental benefits, as it reduces the environmental impacts caused by malpractice related to drug disposal	251 (50.0)
Reusing “unused medicines” is safe, but provided that the unused medicines are stored well	256 (51.0)
Reusing “unused medicines” is safe, but only as long as the unused medicines are of good quality	147 (29.3)
Reuse of “unused medications” is safe, but provided that the quality of “unused medications” is evaluated by a pharmacist or a doctor	273 (54.8)

**Table 5 healthcare-11-00075-t005:** Assessment of factors associated with willingness to reuse unused stored medicines.

Predictor	Willingness to Reuse Unused Stored Medicines			
	[0: Unwilling, 1: Willing]			
	Odds Ratio	*p*-Value ^#^	Odds Ratio	*p*-Value ^$^
Age				
≤40 years	Reference			
>40 years	0.869	0.395	----	----
Gender				
Male	Reference			
Female	0.658	0.026 ^^^	0.616	0.013 *
Marital status				
Married	Reference			
Others (single, widowed, or divorced)	1.553	0.020 ^^^	0.995	0.983
Level of education				
High school or less	Reference			
Diploma or Bachelor’s degree	0.620	0.302	----	----
Postgraduate studies (Master’s or Ph.D.)	0.630	0.335	----	----
Profession				
Non-medical profession	Reference			
Medical profession ^1^	1.863	0.001 ^^^	1.640	0.011 *
Income level				
<500 Jordanian dinars ^2^	Reference			
≥500 Jordanian dinars	0.449	<0.001 ^^^	0.451	0.002 *
Insurance				
Not insured	Reference			
Insured	0.679	0.038 ^^^	1.136	0.583
Are you currently taking medication for a long time (more than six months)?				
No	Reference			
Yes	0.693	0.053 ^^^	0.752	0.199

^#^ Using simple logistic regression, ^$^ using multiple logistic regression, ^^^ eligible for entry in multiple logistic regression, * significant at 0.05 significance level. ^1^: e.g., physician, dentist, pharmacist, nurse, ^2^: Jordanian Dinars = 1.41 United States Dollar.

## Data Availability

The data presented in this study are available on request from the corresponding author.

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
