# Peer review of "Factors Influencing Public Willingness to Reuse the Unused Stored Medications in Jordan: A Cross-Sectional Study"

_healthcare, 2022, doi:10.3390/healthcare11010075_

Round 1

Reviewer 1 Report

Dear Authors,

The aim of the article "Public Willingness to Reuse Unused Stored Medications in the Future and Factors Affecting Their Decisions" is to examine the willingness of the public to reuse unused stored medicines in the future and the factors influencing their decisions.

The work is very well described in all details. The topic is correctly introduced, methods well characterized and the results are clearly presented.

Tables and figures are presented correctly.

The authors point out that using "unused drugs" has economic and environmental benefits. We must ask ourselves what to do with expired drugs.

Therefore, future studies are required to assess healthcare professionals’ perceptions towards medicine reuse and pharmaceutical panties’ willingness to be part or facilitator of medicine reuse.

I think this is an important topic and the authors made an interesting paper, that should be offered to the scientific community. 

With best and warmest regards,

Author Response

Reviewer 1

Dear Authors,

The aim of the article "Public Willingness to Reuse Unused Stored Medications in the Future and Factors Affecting Their Decisions" is to examine the willingness of the public to reuse unused stored medicines in the future and the factors influencing their decisions.

The work is very well described in all details. The topic is correctly introduced, methods well characterized and the results are clearly presented.

Tables and figures are presented correctly.

The authors point out that using "unused drugs" has economic and environmental benefits. We must ask ourselves what to do with expired drugs.

Therefore, future studies are required to assess healthcare professionals’ perceptions towards medicine reuse and pharmaceutical panties’ willingness to be part or facilitator of medicine reuse.

I think this is an important topic and the authors made an interesting paper, that should be offered to the scientific community. 

With best and warmest regards,

Response to reviewer 1

Dear Reviewer 1,

Thank you very much for your comments. We assure you that we will work on exploring healthcare professionals’ perceptions towards medicine reuse and pharmaceutical panties’ willingness to be part or a facilitator of medicine reuse in the future.

Reviewer 2 Report

Medicine waste as a problem has been identified for decades, and there is good evidence that medicine reuse would be a great idea to solve medicine waste. In this manuscript, the authors have investigated the public willingness to reuse unused stored medications in the future and the factors affecting their decisions. The result indicated that the public positively endorsed the idea of reusing unused stored medicines if quality and safety are assured. The question posed by the authors is meaningful, and the study conforms to the scope of the journal; the methods and results are relevant and accurate. Therefore, I think that this manuscript is worth publication.

Author Response

Reviewer 2

Medicine waste as a problem has been identified for decades, and there is good evidence that medicine reuse would be a great idea to solve medicine waste. In this manuscript, the authors have investigated the public willingness to reuse unused stored medications in the future and the factors affecting their decisions. The result indicated that the public positively endorsed the idea of reusing unused stored medicines if quality and safety are assured. The question posed by the authors is meaningful, and the study conforms to the scope of the journal; the methods and results are relevant and accurate. Therefore, I think that this manuscript is worth publication.

Response to reviewer 2

Dear Reviewer 2,

Thank you very much for your comments. We wish that the results from this study impact the community in Jordan and Globally. I hope this article will interest the journal audience, especially considering the need to conduct this study and its impact on tackling the “Medicine Waste” problem by endorsing the idea of reusing unused stored medicines if quality and safety are assured.

Reviewer 3 Report

Dear authors,

Thank you for sharing the intensive work and some results are quite interesting.

I would like to express some suggestions for your consideration.

According to ref. no. 4, the drugs listed are not complete as stated in lines 40-43, "At the community pharmacy, the patient can buy any medicine without a prescription, except for controlled narcotics and major tranquilizers (e.g., benzodiazepines), which can only be dispensed upon the issue of a special prescription signed by a registered physician (4)". Recommend adding more since it would be misleading information on the handling of prescribed medicines.

To further justify the reason that includes only the profession (medical vs non-medical) in Table 3.

In Table 5, change "parameter" to "predictor" for better understanding.

In the discussion, to highlight and further elaborate on the factors i.e. the significant predictors.

The conclusion shall emphasize the prevalence of public willingness to reuse unused stored medications and the important predictors.

Author Response

Reviewer 3

Dear Reviewer 3,

Thank you very much for your valuable comments.

  1. Thank you for sharing the intensive work and some results are quite interesting. I would like to express some suggestions for your consideration.
    According to ref. no. 4, the drugs listed are not complete as stated in lines 40-43, "At the community pharmacy, the patient can buy any medicine without a prescription, except for controlled narcotics and major tranquilizers (e.g., benzodiazepines), which can only be dispensed upon the issue of a special prescription signed by a registered physician (4)". Recommend adding more since it would be misleading information on the handling of prescribed medicines.

Dear Reviewer 3,

Thank you very much for your valuable comments.

Response to reviewer 3, comment 1:

Thank you for your valuable comment. The drugs list is updated and complete now as directed below:

“At the community pharmacy, the patient can buy any medicine without a prescription, except for misoprostol, sedative-hypnotics, controlled narcotics, and major tranquilizers (e.g., benzodiazepines), which can only be dispensed upon the issue of a special prescription signed by a registered physician”.

  1. To further justify the reason that includes only the profession (medical vs non-medical) in Table 3.

Response to reviewer 3, comment 2:

Thank you for your valuable comment. There is no specific reason for only comparing medical vs non-medical individuals in table 3. So, we decided to report only the results of the total population, and to assess the predictors of public willingness to reuse medications in table 5. 

  1. In Table 5, change "parameter" to "predictor" for better understanding.

Response to reviewer 3, comment 3:

Thank you for your valuable comment. Corrected as advised. The word “parameter” is changed to “predictor”.

  1. In the discussion, highlight and further elaborate on the factors i.e. the significant predictors.

Response to reviewer 3, comment 4:

Thank you for your valuable comment. Corrected as advised. Further elaboration on those with a medical professional as significant predictors was added as below:
“Also, those who have medical professions were less willing to reuse unused stored medications. This would be significant considering the need to assess their perception of reusing unused stored medicines and factors that affects their decisions”.

  1. The conclusion shall emphasize the prevalence of public willingness to reuse unused stored medications and the important predictors.

Response to reviewer 3, comment 5:

Clarified and corrected as advised below:

“The prevalence of public willingness to reuse unused stored medications and the important factors affecting their decisions were evidenced in the literature in different countries that reported positive perceptions toward reusing unused stored medicines considering quality and safety assurance.”.

Reviewer 4 Report

I have some comments on your paper titled “Public Willingness to Reuse Unused Stored Medications in the Future and Factors Affecting their Decisions”. Although your paper topic is interesting and this work has a potential to contribute to the literature on health policy and management, many parts of the paper are still weak and the author(s) are highly suggested to revise the paper as much as possible.

Specific comments:

1. The paper title should be revised a bit to make it more compelling.

2. The method section is still weak. Please add/provide a conceptual framework because it can help shape your paper with a strong rationale. More importantly, please provide a “Data analysis” section that describes how you build your regression model and validate it.

3. The discussion section needs further elaboration. Please explain why people make decisions and how they make them. In the light of mindspongecon (Khuc, 2022), people decide to reuse unused stored medications depending upon their core values with different priorities. This is a decision-making process. For example, some may care about the cost-benefit dimensions but others may care about usefulness, preferences, and so on. Besides, the cultural factor may be a key variable that influence the people’s decisions. If you are unable to get information on this factor, you should view this as a paper’s limitation.

4. Please further elaborate the limitation part of the paper as much as possible.

5. There is a key reference that author(s) may use for your revising.

 Khuc, Q. V. (2022). Mindspongeconomics. Working Paper Series No. 2022/9. https://doi.org/10.31219/osf.io/hnucr

Author Response

Reviewer 4

I have some comments on your paper titled “Public Willingness to Reuse Unused Stored Medications in the Future and Factors Affecting their Decisions”. Although your paper topic is interesting and this work has a potential to contribute to the literature on health policy and management, many parts of the paper are still weak and the author(s) are highly suggested to revise the paper as much as possible.

Specific comments:

Dear Reviewer 4,

Thank you very much for your valuable comments.

  1. The paper title should be revised a bit to make it more compelling.

Response to reviewer 4, comment 1:

Thank you for your valuable comment. The title was revised as follows:

“Factors Influencing Public Willingness to Reuse the Unused Stored Medications in Jordan: A cross-sectional Study”

  1. The method section is still weak. Please add/provide a conceptual framework because it can help shape your paper with a strong rationale. More importantly, please provide a “Data analysis” section that describes how you build your regression model and validate it.

Response to reviewer 4, comment 2:

Thank you for your valuable comment. Figure S1 depicts a flowchart of the whole study process is added as supplementary material. Also, the data analysis section was revised as follows:

“Statistical package for social science (SPSS) version 22 (SPSS Inc., Chicago, IL, USA) was used to analyses the data collected in this study. Frequency/percentage was utilized for qualitative variables. The study was conducted to determine predictors of participants’ willingness to reuse the unused stored medications. It was hypothesized that age, gender, marital status, education level, profession, outcome level, the presence of insurance, taking chronic medications could predict public willingness to reuse the stored medications. To test this hypothesis, logistic regression analysis was conducted. Following simple logistic regression, any variable with a P-value < 0.250 was considered eligible for entry in multiple logistic regression analysis. All variables were checked for any absence of multicollinearity before performing multiple logistic regression analysis (i.e., Pearson correlation coefficient <0.9 for any two variables). A P-value (≤ 0.05) was considered statistically significant when identifying factors affecting participants’ willingness to reuse the unused stored medicines. The validity of the multiple logistic regression model was assessed using Nagelkerke R2, and by the application of the Hosmer-Lemeshow goodness-of-fit test with a P-value >0.05 indicated that the model fits with the data.

  1. The discussion section needs further elaboration. Please explain why people make decisions and how they make them. In the light of mindspongecon (Khuc, 2022), people decide to reuse unused stored medications depending upon their core values with different priorities. This is a decision-making process. For example, some may care about the cost-benefit dimensions but others may care about usefulness, preferences, and so on. Besides, the cultural factor may be a key variable that influence the people’s decisions. If you are unable to get information on this factor, you should view this as a paper’s limitation.
  2. Please further elaborate the limitation part of the paper as much as possible.

Response to reviewer 4, comment 3:

Thank you for your valuable comment. This is clarified now as advised and addressed as a limitation of the study as below:

“Despite the positive public willingness to reuse unused stored medicines, it is important to clarify that the public decisions to reuse the unused stored medicines would be affected by factors related to attitudes (e.g., good or bad, harmful or beneficial, and worthless or worthwhile), social norms (e.g., public values, opinions, and thoughts), and control factors (e.g., sealed medicines, medicines quality, and safety checking). This would be another limitation to this study”.

Response to reviewer 4, comment 4:

Thank you for your valuable comment. Limitations were revised as follows:

“The current study should be considered in the light of a number of limitations. First, data were collected online survey; hence, only people who use the Internet and other social media platforms were able to participate. Moreover, all information in this study was obtained through self-report method. There may result in social desirability bias or recall bias. Moreover, we used a convenience sampling methods which is not a representative sample for Jordanians’ population.”

  1. There is a key reference that author(s) may use for your revising.

 Khuc, Q. V. (2022). Mindspongeconomics. Working Paper Series No. 2022/9. https://doi.org/10.31219/osf.io/hnucr

Response to reviewer 4, comment 4:

Thank you for your valuable comment. Cited now.

Round 2

Reviewer 4 Report

Thank you for carefully and appropriately addressing my comments. The revised manuscript is much improved now and I have no further comments on your work.